# Multi-Variable Governance Index Modeling of Government's Policies, Legal and Institutional Strategies, and Management for Climate Compatible and Sustainable Agriculture Development

Kanwar Muhammad Javed Iqbal [1], Nadia Akhtar [2], Sarah Amir [2], Muhammad Irfan Khan [2], Ashfaq Ahmad Shah [3,4], Muhammad Atiq Ur Rehman Tariq [5,6,*] and Wahid Ullah [7]

1  National Institute of Maritime Affairs, Bahria University, Islamabad 44000, Pakistan
2  Department of Environmental Science, International Islamic University, Islamabad 44000, Pakistan
3  Research Center for Environment and Society, Hohai University Nanjing 210098, China
4  School of Public Administration, Hohai University, Nanjing 210098, China
5  College of Engineering, IT & Environment, Charles Darwin University, Darwin, NT 0810, Australia
6  College of Engineering and Science, Victoria University, Melbourne, VIC 8001, Australia
7  Department of Sociology, University of Chakwal, Chakwal 48800, Pakistan
*  Correspondence: atiq.tariq@yahoo.com

**Abstract:** Agriculture has a very strong nexus with water and energy sectors due to their complex interdependence and interplay in the context of adaptation, resilience, mitigation, and low carbon development to cope with the increasing effects of a changing climate. The situation demands a comprehensive response in terms of policies, legal instruments, institutional strategies, and management in the national, sub-national, and local contexts of the governance for climate compatibility, environmental security, and sustainable agriculture development; particularly in developing countries, as they are at the forefront of a high vulnerability risk and severe environmental insecurity due to a business-as-usual complex and weak governance. Therefore, the present study aimed to assess the adequacy of the climate response vis-à-vis policies, legal and other appropriate arrangements in place for agriculture governance by reviewing the high vulnerability case of Pakistan. Considering the need, the assessment model was developed using the first climate principle, nine criteria, and 43 composite indicators. A multi-criteria decision analysis method along with Simple Multi Attribute Rating Technique (SMART) on a ratio scale, combining qualitative and quantitative data and employing rule-based and rights-based governance approaches were adopted to collect and analyze a dataset of 357 observations from 17 locations, which were validated through Pearson Correlation, Regression, and KW H-Tests. The findings reveal significant gaps at the federal, provincial, and district levels in policies, legal and institutional strategies to step forward the climate agenda in Pakistan's agriculture sector. However, the inadequacy is not homogenous at all levels of governance. The overall situation is similar to what has been reported in developing countries in the United Nations Report on Sustainable Development Goals (SDGs) 2020. Provincial climate strategies are required along with enhanced coordination and capacities for execution at all tiers of constituencies.

**Keywords:** multi-variable modeling; governance index; agriculture; policy and strategy; climate compatible development; MCDA

## 1. Introduction

Effective policy and planning processes are the key to progress, and perform a pivotal role in the sustainable development of countries [1]. Underprivileged communities residing in less-developed countries are at greater risk of adverse impacts arising from incoherent policies, and planning which consequently destabilize the development process and accelerate the degradation of natural resources. Addressing environmental challenges in sectoral policies and their implementation across all sectors is a key to sustainable development [2].

Traditional governance framework with segmented responsibilities impedes assigning distinct responsibilities to various sectors. Thus, segmented governance models, coupled with the absence of coordination among sectors, become a major hurdle to resolving complex problems, such as climate change. Distortions in the governance framework are deep-rooted due to differential and non-compromising standpoints of stakeholders augmented with an incompatibility of interests. Consequently, achieving a level of trade-off in policy-making becomes nearly impossible [3]. Worldwide, governments are making substantial adjustments in traditional institutional and governance structures, however, mechanisms for addressing cross-sectoral and global issues, such as climate change, frequently demonstrate a dearth of coherence and integration in sectoral policies. Climate resilience requires horizontal and vertical coherence among policies and coordination among institutions [4], thus, posing a serious challenge to the traditional governance system.

Scientific evidence proves that the global temperature has been rising for more than a hundred years, and climate change threats are now visible across the globe (in natural and human systems) [5]. During the last decade, the changing climate and its associated impacts have been known as the foremost scientific, political, economic, and environmental issues [5–8]. Weather and climatic disruptions are triggering desertification [9,10], causing glacial melting [5], unleashing river/coastal flooding [11], and squeezing agricultural productivities [12,13]. Scientists predict that an increase in carbon dioxide concentration is elevating the rate of photosynthesis and consequently plant growth and productivity; at the same time, high temperatures have increased plant respiration and evapotranspiration rates, resulting in reduced crop productivity. Changing climates also result in higher pest infestation, shifting patterns of weed flora, and condensed crop duration [14].

Thus, cumulative impacts of climate change are a serious risk for environmental security and sustainable development considering the strong nexus of agriculture with water and energy sectors vis-à-vis adaptation, resilience, mitigation, and low carbon future. On the one hand, greenhouse gas (GHG) emissions from point and non-point sources in agriculture are an important segment that requires climate-smart practices at the farm level. It is a critical aspect to meet 2030 targets under the SDG-13 and Paris Climate Agreement 2015, which are lagging behind as reported in United Nations SDG Report 2020 [15]. While on the other hand, it is evident that climate change is stressing food production, which eventually escalates food prices and potentially threatens food security. The demand for food is anticipated to increase by about 300 percent by 2080. If, as expected, food production declines due to global warming, there is likely to be further pressure on food prices, increasing the current threats to food security [6].

Both the global and national security paradigms are at great risk due to the devastating impacts of climate-induced anomalies, if required measures to mitigate and adapt to such changes are not taken both by developed and developing countries [16]. The long-term challenge faced by policymakers and planners in the agriculture sector is to maintain the productivity of the land by efficiently, profitably, and safely producing food and cash crops to fulfill the increasing demand of the world population. Developing countries are facing difficulties in embracing this challenge, being more vulnerable to climate change due to their poor resource-base, technical and financial constraints, inconsistent and incoherent policies, their legal and institutional framework, and their inadequate adaptive capacity. Their resilient capacity is also low due to their reliance on climate-stressed water resources, such as agriculture in Africa and South Asia [17], and Pakistan is no exception [18]. According to global assessments, an overall mean crop yield declined by 8% in South Asia, with pronounced reductions projected for major crops i.e., "wheat (12%)", "maize (7%)", "sorghum (3%)" and "millet (9%)" [17]. Consequently, developing countries, such as Pakistan, with inadequate access to innovative technology and agricultural knowledge, are struggling hard to cope with climate stress [13,19]. As a whole, the cross-sector nexus of agriculture has a complex interdependence and interplay. The situation demands a comprehensive response in terms of policies, legal instruments, institutional strategies, and management arrangements in the national, sub-national, and local contexts of the

governance for climate compatibility, environmental security, and sustainable agriculture development; particularly in developing countries as they are at the forefront of high vulnerability risk and severe environmental insecurity due to a business-as-usual complex and weak governance. In this context, the case of Pakistan has a high level of relevance due to its presence among the top 10 globally ranked vulnerable countries worldwide, while the adequacy of the state of climate governance has many queries vis-à-vis the size of the country's agriculture-based economy, the supply and demand nexus, and international obligations committed through its statement regarding nationally determined contributions (NDC) [20–23].

Climate change is causing serious harm to Pakistan's agriculture with its adverse impact on society and the economy [24]. Pakistan is an agro-based country, with a high vulnerability to climate change due to climate-sensitive agricultural practices. Crops, particularly cereals, are vulnerable to increased temperature in Pakistan. Estimates show that a 1 °C rise in temperature can cause a 5–7 percent decline in wheat yield [25]. Researchers estimate a 6–9 percent decline in wheat production in Pakistan's sub-humid, semi-arid, and arid regions, while it could rise in the wet zone. However, impacts are not uniformly distributed [26]. For example, a temperature rise of 1.5–3 °C will decrease wheat production by 7% to 21%, respectively, in Swat, and will cause an increase by 14–23% in the Chitral district of Pakistan. Contrary to that, agriculture policies have failed to address climate change challenges. The agriculture policy drafted in 1991 is the only guiding document, but it does not discuss climate change as a potential threat. In many instances, proposed policy measures contradict the climate change response strategies, e.g., the provision of a subsidy on tube wells in Balochistan is counterproductive for climate change. The complexity of climate change and the agriculture sector demands an inclusive and comprehensive policy for climate-compatible development (CCD). However, the adaptive capacity of Pakistan is limited, due to a lack of resources, though adaptation projects for different provinces in Pakistan are being executed with the support of international organizations [27].

Appreciably, Pakistan has played a proactive role in global climate negotiations, which is expressed in the form of the "National Climate Change Policy (2012)" and the "Climate Change Policy Implementation Framework (2014–2030)", and the revised version of the climate change policy in 2021 for the successful implementation of climate change issues through strategic planning. "Pakistan Climate Change Act 2017" has also been passed to materialize measures to combat climate change. The "Climate Change Fund", the "Climate Change Council" and the "Climate Change Authority" were established by the legislation to enact the Climate Change Act [28]. However, aligning climate change response strategies with sectoral policies is far from a satisfactory level [22,29]. There is a query arising of whether the existing responsibility for the first governance component is inclusive and adequate to meet the CCD agenda in the agriculture sector, which is the basis for the null hypothesis that there is no such mechanism so far established or in existence. The overall subject context needs to be assessed for the identification of the gaps and determining the actual agenda for CCD against the basic response mechanism by employing a suitable methodological framework. Although the subject of governance resonates well in the existing literature [30,31], there are limitations in the available methodological frameworks [32–34] for the assessment of multi-variable governance cases, due to the involvement of diverse and multi-dimensional concepts, approaches, principles, criteria, and indicators to address all aspects of CCD [35–40]. The application of a single approach or method would not serve the need. Therefore, the present study aimed at multi-variable governance index modeling with an innovative and integrated approach to assess the adequacy of climate response vis-à-vis the government's policies, legal instruments, institutional strategies, and management arrangements in place for the agriculture sector, by taking the high vulnerability case of Pakistan. The study employed principles (P), criteria (C), and indicators (I) for assessing the climate response mechanism and institutional arrangements for climate compatibility, environmental security, and sustainable agriculture development,

with an auxiliary objective to replicate the model framework in other developing countries through knowledge addition.

## 2. Methodological Framework

### 2.1. Study Approach, Variables, and Design

This paper is extracted from a broad Ph.D. research study done by the lead author, by reviewing three important sectors i.e., agriculture, water, and energy, which have a very strong nexus in the context of CCD. Considering the limitations of available methodologies, an integrated approach through the engagement of the expert group was employed for the development of a multi-variable model framework comprising principles, criteria, and indicators (PCI) to assess the compatibility and adequacy of the existing governance instruments for CCD. The logical sequence adopted for the study is based on two steps. In the first step, the PCI framework was developed, while in the second step, the framework was applied to the case of the agriculture sector of Pakistan. The application of the framework comprehended federal, provincial, as well as district levels. The study exercised a combination of governance approaches (rule-based and rights-based) [41–45]. For this purpose, three consultation meetings were organized to obtain input and build a consensus on the PCI framework. The expert group comprised academics and subject professionals in Islamabad. The initial findings through a qualitative content analysis of available literature vis-à-vis reported PCIs were used for brainstorming purposes prior to concluding and consolidating the PCIs for CCD in such a way to utilize them fully or partially for different components and tiers of the governance and constituencies, respectively. The brainstorming session employed the situational/problem tree analysis technique for rationalizing the different climate scenarios. The situational/problem tree analysis is a well-established research strategy for effective decision-making in such scenarios [46–52]. These practices are widely used for good planning and management cycles through a cause-and-effect analysis, which can be easily produced through focus group discussions (FGDs) [46]. As an outcome, six novel principles for climate response i.e., CPs were settled for six governance components (See File S1; Supplementary Material). Subsequently, nine criteria were finalized. These components are mutually connected to provide a comprehensive assessment of the governance framework as well as being self-governing to be used as a single component vis-a-vis a single principle. The six novel climate principles (CPs) and nine CCD response criteria (as shown in Figure 1 and Table 2) are unique in the sense that they are specific to the governance of the CCD agenda and generic by means of their application to all sectors with full or partial application [49,53]. Six climate principles correspond to the respective six components of the basic governance mechanism (Figure 1), while the set of nine criteria is applicable to all six climate principles (CPs) and components of basic governance i.e., GC1 to GC6. The only sector and governance component specific items are the set of composite indicators against each criterion, which integrates the scope of climate principles, good governance principles, aspects of different rule-based and rights-oriented approaches of governance, and measures for CCD response strategies, which is why the criteria are termed as CCD response criteria. It is anticipated that the derived six climate governance principles (CPs) will act as the main vehicles and nine criteria will be the precursors for CCD to carry forward the agenda in all sectoral economies. Sector-specific indicators will be the means of verification for that particular segment of the sectoral economies per se, to assess the adequacy of the overall governance framework for climate compatibility, environmental security, and sustainability.

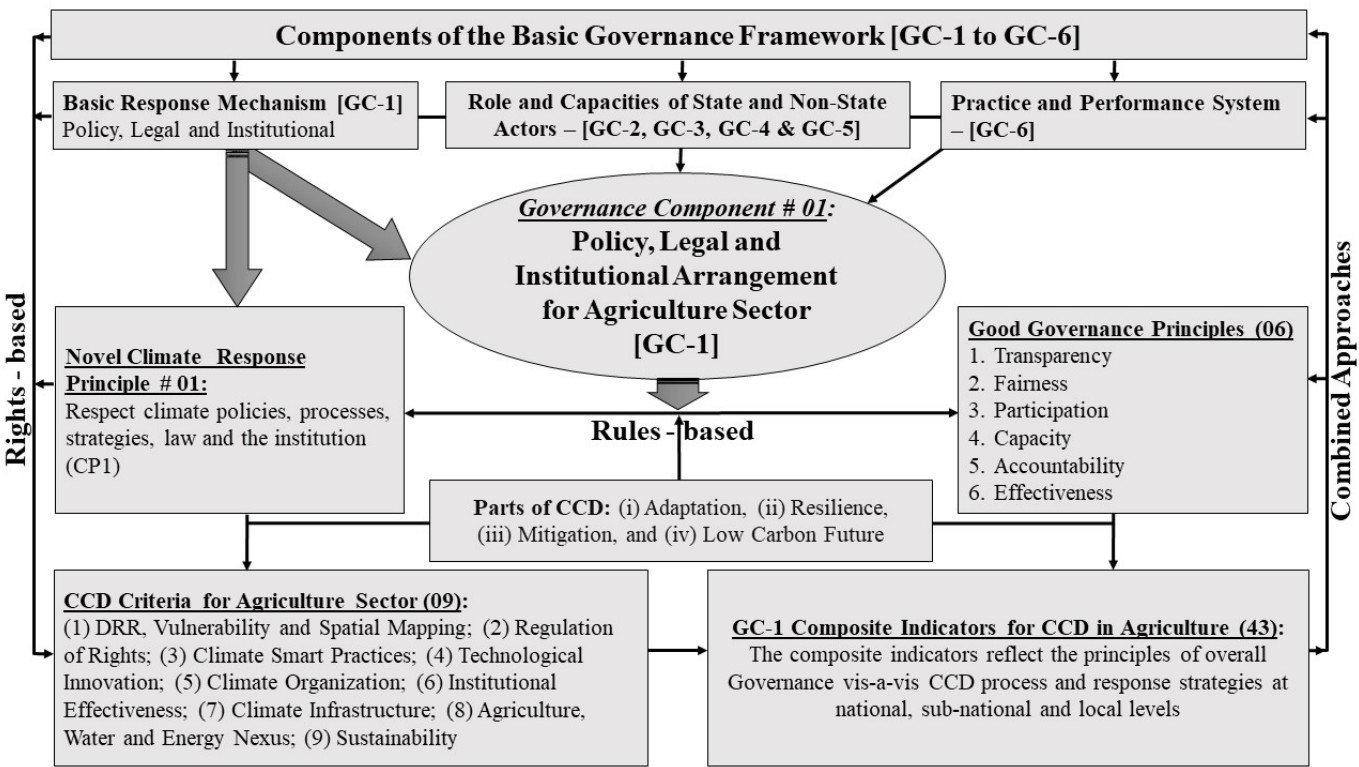

**Figure 1.** Principles, Criteria, and Indicators (PCI) based multivariate model adopted for the study.

In the overall methodological context, this paper is developed based on the first climate principle (CP-1) vis-à-vis its respective first governance component (GC-1). For this purpose, a set of 43 composite indicators was determined against nine CCD response criteria, keeping in view all four parts of the CCD conceptual framework i.e., adaptation, resilience, mitigation, and low carbon future. Figure 1 portrays the multi-variable analysis model design adopted for the study while the next section provides detail about the scope of each CCD criterion for determining the indicators (File S1).

### 2.2. CCD Criteria-Based Scope of Indicators for GC1 in Agriculture

The foremost important criterion is based on the response measures for the 'disaster risk reduction' in the agriculture sector including the 'vulnerability and spatial mapping', which comes under the scope of adaptation and resilience segments for the conceptual framework of CCD. With four indicators, it deals with the in-practice well-established measures and provisions in strategies, policies, and legal institutional mechanisms under the first governance component (GC1) to govern, regulate and manage the vulnerability assessment and its spatial mapping, Local Adaptation Plans of Actions (LAPAs), responding to climatic extreme events, and having a vibrant early warning system that is relevant for the agricultural sector in disaster risk reduction. Geographical knowledge about the vulnerability scale of different areas is critically important to considering the diversity of the ecosystem in developing countries. In addition, capacity building of all relevant actors, including farmers and local communities, and redressal mechanisms through proper handling of grievances were considered and included in the context of policies, legal instruments, and courses of action by the relevant institutions at all tiers of the constituencies in the context of national, sub-national and local climate scenarios.

The second criterion, regarding 'regulation of rights' for the integrated rule-based and rights-oriented approach in the agriculture sector, is critically important for a climate response considering needs due to prevailing diverse concepts and approaches of governance i.e., from informal to formal and rule-based to rights-oriented. It has high significance for the dry river belt areas where encroachments and unattended settlements vis-à-vis

agricultural activities exist which are not only important for economic losses but also put at stake the life and health of human beings in case of extreme climatic events, thus requiring climate adaption and resilience. It also has external linkages with GHG emission factors particularly due to agricultural intensification and fuel energy consumption in agricultural machinery thus becoming relevant for mitigation and a low carbon future. Specific to the scope of GC1, with three indicators, this criterion deals with the existence of well-established measures and provisions in policies, legal instruments, strategies, and institutional arrangements to govern, regulate and manage agricultural rights vis-à-vis diverse geographical areas; particularly restricting the rights for agricultural intensification in mountainous slope and dry river belt areas, which have a higher degree of susceptibility to climatic variations. In this context, it also addresses the need for the capacity development of relevant stakeholders, including the local communities.

The third criterion, i.e., 'climate smart practices' for environmental security and sustainability in agriculture, encompasses all four parts of the CCD conceptual framework by dealing with the existence of established measures and provisions in policies, legal instruments, strategies, and other relevant institutional arrangements to govern, regulate and manage the climate-smart agriculture practices for the promotion of CCD agenda. Mechanisms requiring climate-smart practices, risk recovery approaches for the management of crops' failures, knowledge-based capacity enhancement of all relevant actors, creating mass awareness, and measures for minimizing GHG emissions from livestock and agriculture sources are also covered. A set of five indicators was determined and used for this criterion.

The fourth criterion, i.e., 'technological innovation' in agriculture for climate compatibility, environmental security, and sustainability, deals with the measures and provisions in policies, legal instruments, strategies, and other relevant institutional arrangements to govern and manage the contour farming, seed development, climate resilient breeds, biotechnological solutions, and new conservation methods, etc. by promotion of research and development on CCD, along with best on-farm management and agricultural practices. It also covers the placement of a code of conduct to deal with the subject of genetically modified varieties of the crops vis-à-vis resistance to emerging pest dynamics in changing climatic scenarios. In this context, capacity enhancement of all relevant actors remains very much central to the subject because research-oriented technological solutions have not only technical aspects but also involve legal and ethical codes. It is covered in the context of best practices for drought-related solutions vis-à-vis reclamation of the vulnerable agricultural land. Four indicators were determined and used for this criterion.

The fifth criterion, i.e., 'climate organization' for well-preparedness in the agriculture sector, covers all parts of the CCD conceptual framework by dealing with the existence of established measures and provisions in policies, legal instruments, strategies, and other relevant institutional arrangements to govern, regulate and ensure accountability, the need for legal enforcement and implementation of CCD agenda. It also addresses the synchronization of institutional arrangements at various levels, along with the integrity and capacity of senior-level relevant departmental posts dealing with the CCD agenda in the sector. Furthermore, the development of mechanisms and rules for disaster risk reduction as early warning systems is also considered. Five indicators were determined and used for this criterion.

The sixth criterion, 'institutional effectiveness' for all four parts of the CCD agenda in the agriculture sector deals with the existence of established measures and provisions in policies, legal instruments, strategies, and other relevant transparent institutional arrangements to govern, regulate and ensure a well prepared, properly coordinated and vibrant response mechanism at all tiers of the constituencies in the national, sub-national, and local contexts. It also calls for effective coordination between local (district), sub-national (provincial), and national (federal) line departments for capacity building, guaranteeing inclusive decision-making, enforcement of a state-of-the-art monitoring system in agriculture sector governance for CCD, with a bottom-up philosophy. Additionally, the need for mechanisms and rules for early warning systems for disaster risk reduction, along with

procedures to halt any fraudulent action or corruption in ensuring compliant management in the sector is covered. Eight indicators were determined and used for this criterion.

The seventh criterion, i.e., 'climate infrastructure' for CCD in the agriculture sector, deals with the existence of established measures and provisions in policies, legal instruments, strategies, and other relevant institutional arrangements to govern, regulate and manage the infrastructural needs for research-based technological solutions, need for storage and processing of products, and awareness-raising endeavors. It also discusses the formulation, development, and intermittent review of strategies, policies, and regulations related to the agriculture sector that simultaneously meet the requirements of the CCD agenda. Six indicators were determined and used for this criterion.

The eighth criterion i.e., the agriculture, water, and energy nexus is also very important for all four parts of the CCD conceptual framework considering the cascading effects and interplay due to climatic variations. It deals with the existence of established measures and provisions in policies, legal instruments, strategies, and other relevant institutional arrangements to govern, regulate and manage effective coordination through vibrant departmental linkages for CCD, particularly addressing the complex interdependence and interplay for the nexus of agriculture, water and energy sectors at a national level in general, and local and sub-national levels in particular. Five indicators were determined and used for this criterion.

The ninth criterion, i.e., 'sustainability' for the agriculture sector is a dependent variable on all the above eight criteria. It deals with the existence of established measures and provisions in policies, legal instruments, strategies, and other relevant institutional arrangements to govern require, regulate and manage the environmental safeguards and security related to water security/conservation considering the extreme climatic events, reduction of GHGs in the overall value chain and organic farming, etc., for CCD agenda. It also addresses the social safeguards relating to poverty, farmers' livelihood, and food security, etc. Similarly, economic security is also addressed in relation to the impacts on the conservation of input resources, low-cost technologies, and GDP, etc. for the CCD agenda in the agriculture sector. Three indicators were determined and used for this criterion.

### 2.3. Data Collection

The study is based on an assorted set of variables to apply climate principle 1 against component 1 of the governance framework (GC-1) based governance model in the agriculture sector. Consultative meetings, as described under Section 2.1 above, helped in confining up to nine criteria and 43 indicators for legal, policy, and institutional arrangements (GC-1) [54] (see Figure 1).

For the weighting and scoring of the indicators against each criterion, the "Simple Multi-attribute Rating Technique (SMART)" of MCDA ("Multi-Criteria Decision Analysis") method was used. The ratio scale was adopted with a range of scoring from 0 to 10 for weighting by the respondents i.e., no response/not applicable (0), very poor (0.01–1.99), poor (2.00–3.99), considerable response (4.00–4.99), fair response (5.00–5.99), good progress (6.00–7.49), very good performance (7.50–8.99), and excellent achievement (9.00–10.0). MCDA's SMART is a very effective technique to produce quantitative indices for different issues including the governance aspects to help the decision-making process at all levels, which is why it is well-established with wide recognition and practice worldwide [49,55–58]. A structured questionnaire based on a scoring matrix was developed using the SMART ratio scale for the set of 43 indicators of agriculture governance for Climate Principle 1 titled '*respect climate policies, processes, strategies, law and the institution*' (File S1 for a detailed set of six principles; Supplementary Material). Pilot testing to weigh, normalize and validate the indicators was carried out in Islamabad. Responses were collected from key informants working in agriculture and related federal, provincial, and district departments. The sampling plan was designed based on the geographical boundaries of federal, provincial, and district departments, but at the same, it was limited for the sample size. For this purpose, responses were collected from seven federal and provincial capitals

(i.e., Islamabad, Peshawar, Lahore, Karachi, Quetta, Gilgit, and Muzaffarabad) along with ten districts as shown in Table 1 for sub-national scenarios.

**Table 1.** Selected districts for study.

| Scheme | Provinces (Sub-National) | Districts (Local Context) |
|:---:|:---:|:---:|
| 1. | KPK | Swat, Mansehra |
| 2. | Punjab | Bahwalpur, Rajanpur |
| 3. | Sindh | Badin, Sanghar |
| 4. | Balochistan | Khuzdar, Jhal Magsi |
| 5. | AJK | Muzaffarabad |
| 6. | Gilgit Baltistan | Ghizer |

These districts were selected based on previous or ongoing projects and programs on climate change mitigation, adaptation, or assessing vulnerabilities by the government, non-governmental organizations (NGOs), academic institutions, civil society organizations, and the private sector. A total of 357 responses were collected: one FGD and twenty KIIs (key informant interviews) per sampling location.

*2.4. Data Analysis*

Responses collected were tabulated and cleaned in MS Excel 2016. The criteria-wise and constituency-wise governance index was calculated by averaging the individual score against the relevant indicator. Validation of results was carried out through the "Kruskal–Wallis hypothesis test (H-test)", "Pearson Correlation" and "Regression". For this purpose, the Statistical Package for Social Sciences (SPSS) 25 was used. Before designating whether the samples were statistically dominating for different governance levels, the H-test facilitated the identification and distinguishment of the sample groups and variables in relation to the constituency and the gender. It validated the originality of the data and normalization. One-tailed Pearson correlation analysis facilitated the analysis of the relationship, the impacts, and interlocking of variables. The third test i.e., Multivariate Linear Regression analysis was used to interpret the association between different interconnecting variables to answer the research question.

## 3. Results

The findings of the study are tabulated in the form of a governance index for each criterion as well as each constituency (Table 2). The overall average index score remained at 5.35, depicting a 'considerable' index score on the ratio scale. A constituency-wise comparison revealed a better index score at the federal level (7.52), while a poor (3.93) score was revealed at the district levels. The criteria-wise highest average index score was calculated for criterion six i.e., institutional effectiveness. The lowest index score was calculated for criterion nine i.e., sustainability. Figure 2 represents an overall average governance index for nine criteria for federal, provincial, and district constituencies while Figure 3 shows a cluster bar chart for criteria-wise index scores for all four provinces, AJK, GB, and federal areas. The lowest index score was calculated at the district level however a district-wise comparison showed variations. Figure 4 portrays the GC-1 index at the district level; the highest index score (5.08) was for district Bahawalpur (province Punjab) while lowest index score (2.29) was for district Jhal Magsi (province Balochistan).

**Table 2.** Constituency-wise Breakdown of the Index Score.

| Criteria | Criteria-Wise Index Score | | | | |
|---|---|---|---|---|---|
| | Federal Level | Provincial Level | District Level | Average Index Score | Ranking |
| AC-1.1: Disaster Risk Reduction, Vulnerability and Spatial Planning | 9.12 | 5.92 | 4.80 | 6.61 | Good |
| AC-2.1: Regulation of Rights | 7.14 | 3.02 | 2.54 | 4.23 | Considerable |
| AC-3.1: Climate-Smart Practices | 8.92 | 4.84 | 4.38 | 6.05 | Good |
| AC-4.1: Technological Innovation | 7.39 | 4.61 | 4.19 | 5.40 | Fair |
| AC-5.1: Climate Organization | 7.86 | 5.01 | 4.36 | 5.74 | Fair |
| AC-6.1: Institutional Effectiveness | 8.89 | 6.02 | 5.21 | 6.71 | Good |
| AC-7.1: Climate Infrastructure | 5.71 | 4.40 | 3.96 | 4.69 | Considerable |
| AC-8.1: Agriculture, Water and Energy Nexus | 7.36 | 4.69 | 4.02 | 5.36 | Fair |
| AC-9.1: Sustainability | 5.30 | 2.90 | 1.91 | 3.37 | Poor |
| Overall Average | 7.52 | 4.60 | 3.93 | 5.35 | Fair |
| Ranking | Very Good | Considerable | Poor | Fair | - |

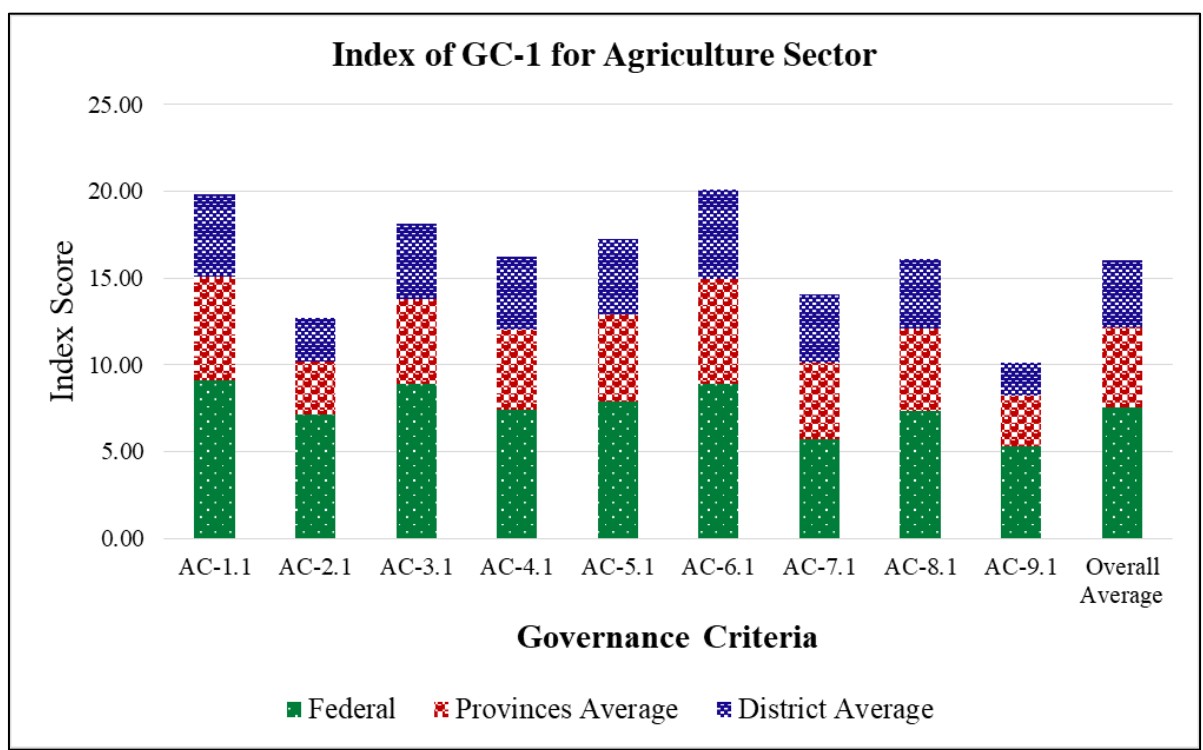

**Figure 2.** Constituency-wise comparison of index score for each criterion (AC-1.1 to AC-9.1).

Findings of statistical validation (KW Hypothesis Tests) based on constituency and gender were analyzed (File S2). Results are presented in terms of asymptotic significances with their respective significance level (0.05) against a total of 357 samples, thus rejecting the null hypothesis. The results of the analysis validate the observations and represent diverse responses from all constituency levels.

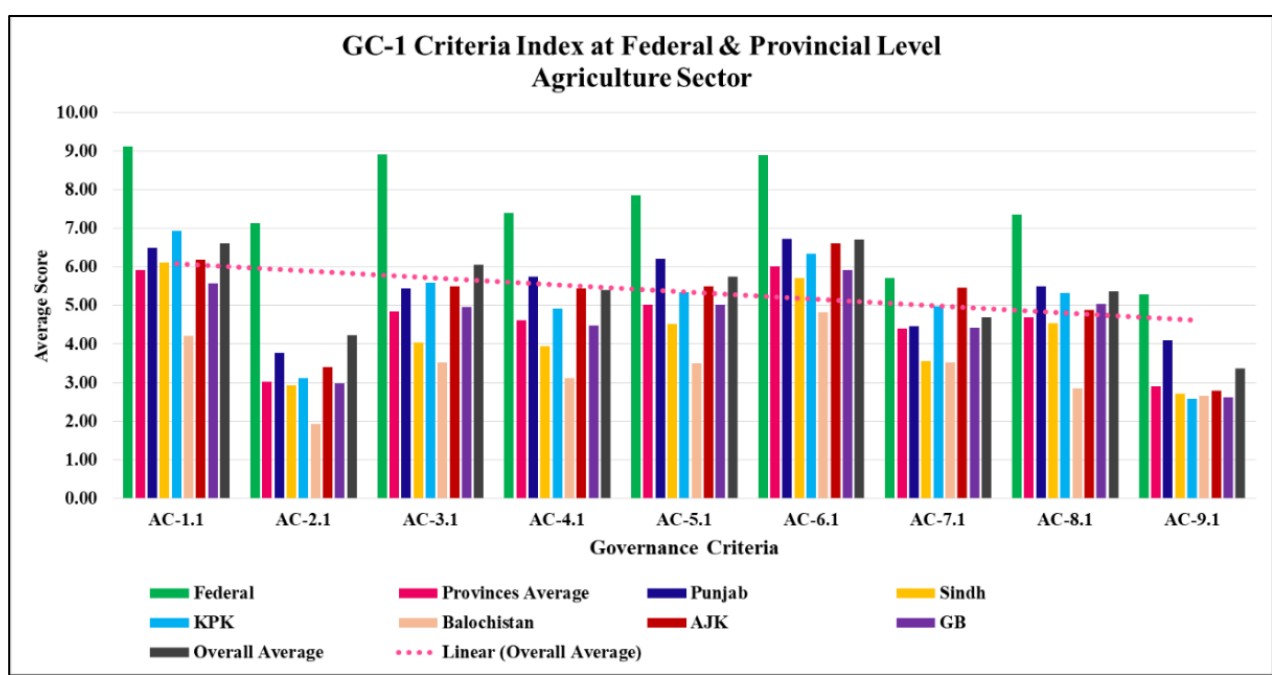

**Figure 3.** Agriculture governance index for governance component 1 (i.e., basic response mechanism)—Criteria-wise (AC-1.1 to AC-9.1).

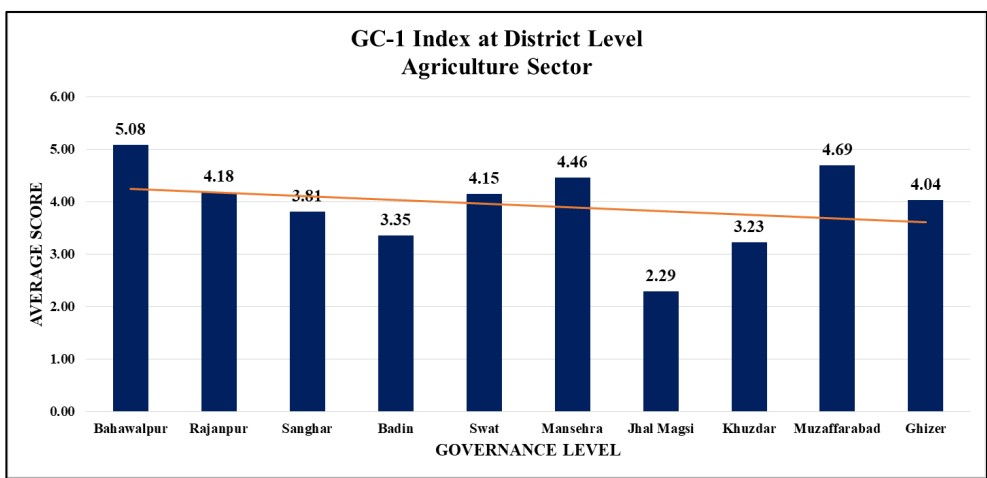

**Figure 4.** Average Index Score at Districts Level.

A strong correlation (1-tailed significance of 0.01) exists between all CCD criteria under GC-1; the summary is shown in Figure 5 and detailed SPSS output is given in File S2. However, the criterion for climate infrastructure (AC-7.1) has a weak correlation with the criteria for agriculture, water, and energy nexus (0.694) and regulation of rights (0.561). AC-7.1 has a very weak correlation (0.447) with AC-9.1 (sustainability). Multivariate regression analysis of the agriculture sector was carried out (SPSS output in File S2, Supplementary Material). For this purpose, the sustainability of the governance framework i.e., policies, laws, and institutions, is considered to be the dependent variable. For the linear regression model, values of R (0.915) and R Square (0.837) showed good variance. The *t*-test showed a significant relationship between the criterion of sustainability (AC 9.1) and the other eight criteria (above ±2) except the criterion of technological innovation (AC 4.1) for which the value was 1.561. Collinearity diagnostics (tolerance > 0.10 and VIF < 10) for criteria of climate-smart agricultural practices (AC-3.1) and climate organization (AC-5.1) did not show significance; all criteria except the criterion of climate infrastructure (AC-7.1) showed

good zero-order correlations with the criterion of sustainability (AC-9.1). The normal P-P plot (Figure 6) presented results with little upward and downward fluctuations. The scatter plot (Figure 7) showed three, different groups, within the ±3 boundaries, except a very few outliers.

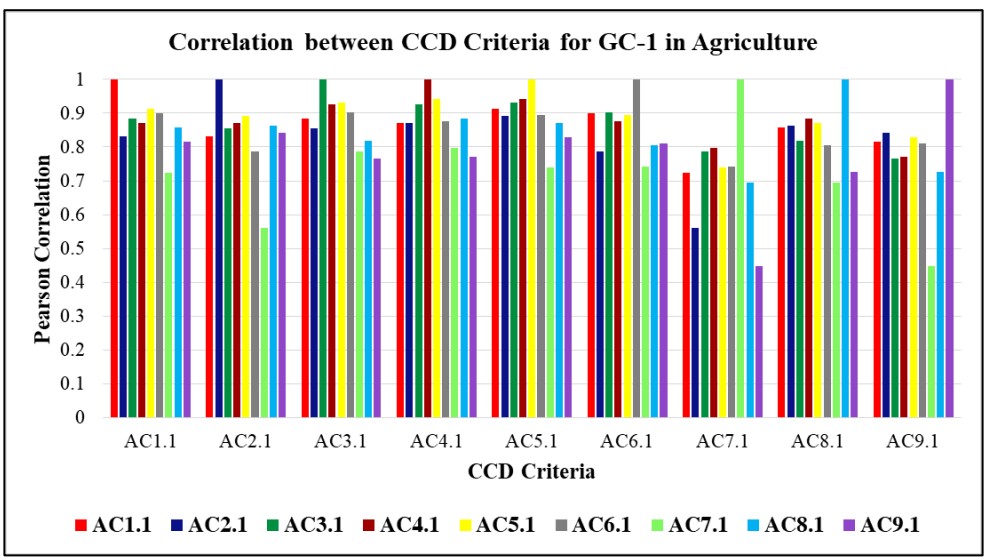

**Figure 5.** Criteria-wise (AC-1.1 to AC-9.1) Pearson Correlations.

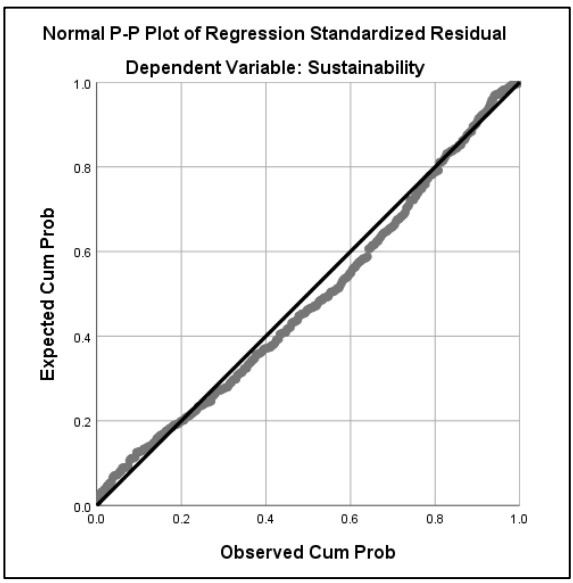

**Figure 6.** Regression standardized residual—P-P plot.

It is deduced that all nine criteria impact each other, however, the basis for the null hypothesis regarding the placement of inclusive and adequate climate response vis-à-vis government's policies, legal instruments, strategies, and other institutional arrangements is that there is no such mechanism so far established or that exists that cannot be rejected for the overall case of the first governance component. It is construed that a coherent and inclusive response mechanism to address climate change impacts in the agriculture sector is absent.

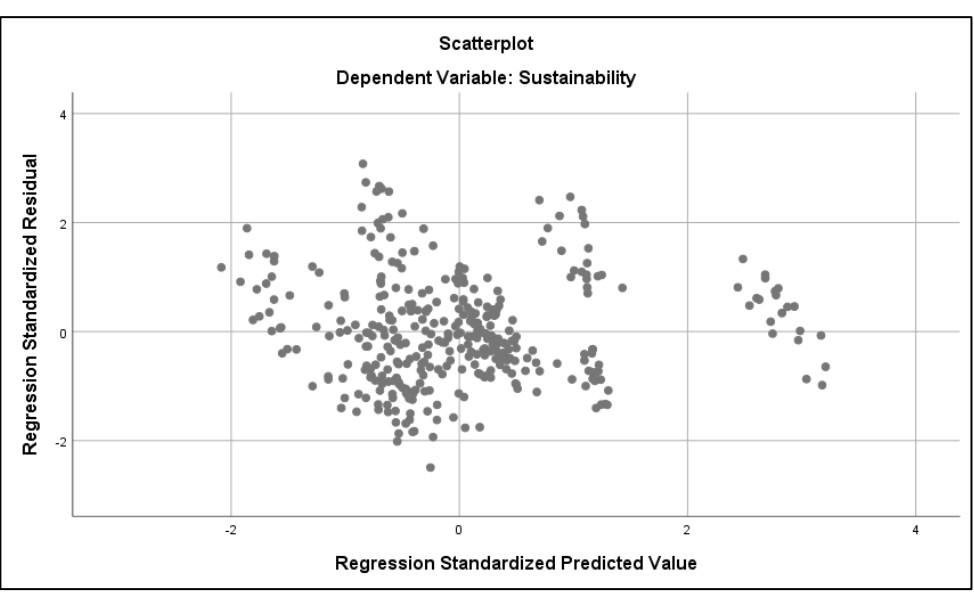

**Figure 7.** Regression standardized residual-scatterplot.

## 4. Discussion

Climate change governance is most relevant to Pakistan, which is located in a geographical region where the impacts of climate change are more evident than in other regions [20,21]. Pakistan is an agrarian economy, with about 60% of the area receiving rainfall of less than 20mm and is classified as arid to semi-arid, consequently, major dependence is on rivers for agriculture [59]. The annual agriculture growth rate (average 4–4.5%) is highly variable with the highest growth rate (11.7%) in 1994–1995, contributed to by a high production of cotton, gram, milk, and meat. Since the green revolution in the 1960s, the agriculture sector in Pakistan has become less diverse i.e., the agricultural economy now depends on a few major crops, such as wheat, maize, cotton, sugarcane, and rice, while other crops contribute only a 10% share of the total production [60]. Crop yields are generally low due to limited access to good quality seeds, inefficient irrigation, lack of technological irrigation, lack of education, and variable climate in Pakistan [61]. Policy measures that are aimed at a "green revolution" have resulted in problems, such as depleting groundwater aquifers, water logging, salinity, and water pollution. These problems are now further augmented by the impacts of climate change on soil, water, land, and weather patterns which ultimately reduce crop productivity [13,19]. Climate change is resulting in erratic monsoon patterns and accelerated glacier melting, which causes floods, shifting cropping seasons, and a decline in crop productivity. The reported projections based on the recent studies infer that the scale and speed of such incidents will increase in the future [18], however, effective strategies can moderate or lessen these impacts. This requires an inclusive, and climate-compatible policy, and legal and institutional framework.

Contrary to that, work on the development of a comprehensive policy for agriculture has remained slow in Pakistan. Achieving food self-sufficiency has remained the first priority of the agriculture policy since 1980, which was the inception of the first agriculture policy in Pakistan. The National Agriculture Policy of 1991 reiterated these priorities, and the same was supported by several other policies addressing different aspects of the agriculture sector e.g., "Corporate Farming Policy", "Wheat Policy", and "Cotton Policy" were also in place. Besides that, "Agriculture Perspective and Policy" [62], "National Medium Term Development Framework" [63], and "Vision 2030" [64] were also serving as policy documents. In 2009, a "Draft National Agriculture Policy (2009–2010)", was prepared with the technical assistance of the Asian Development Bank (ADB). However, the policy could not get approval due to the 18th Amendment of the Constitution of 1973, which resulted in the devolution of the agriculture sector from federal to provincial. The draft policy

claimed a strong footing on its policy measures in "MTDF (2005–2010)" and "Vision 2030". NAP (2009–2010) proposed measures to improve efficiency, safeguard sustainability, and boost competitiveness. The policy proposed an enabling framework for enhancing crop production by positively supporting food security; improving productivity; crop diversification; management of land and water resources; climate change adaptation; disaster risk assessment; and agriculture extension services. The policy also addressed the challenges linked to disasters and climate change. However, response strategies were limited to assessing threats, including the assessment of vulnerability and disaster preparedness without identifying the need for capacity building of smallholding farmers as well as institutions.

However, in the milieu of climate vulnerability of agriculture, the adoption of approaches that ensure climate resilience is instrumental to significantly contributing toward climate mitigation and adaptation as discussed in the Framework for Implementation of Climate Change Policy (FICCP). Establishing suitable connections between functionally interconnected issues boosts the prospects for problem-solving and improves the efficiency and effectiveness of policies [65]. For example, FICCP has strong linkages with the agriculture sector policies and legal and institutional arrangements but the same needs to be integrated at the provincial and district levels of decision making as agriculture is a provincial subject after the 18th Amendment [53]. Thus, the assessment framework developed in the present study is built on the premise that climate change and agriculture have cross-sectoral linkages and overlaps for policies, strategies, and legal and institutional mechanisms which require horizontal as well as vertical coherence [6,65,66]. Therefore, a comprehensive and inclusive governance framework should be based on the policies and legal and institutional arrangements that can address climate change challenges. Therefore, nine criteria (AC 1.1–AC 9.1) were opted to keep in view the functional linkages of the climate and agriculture sector i.e., disaster preparedness, climate change mitigation, vulnerability, adaptation, and resilience.

Disaster preparedness (AC 1.1) assessed GC-1 based on indicators of vulnerability assessment, including spatial mapping, Local Adaptation Plans of Actions (LAPAs), reducing GHG emissions, and an early warning system for disaster preparedness. The findings revealed that the governance components against the criteria are comprehensive and inclusive (Index Score 6.61) covering all four aspects as identified in the indicators of AC 1.1. The findings corroborate the fact that federal and provincial disaster risk management plans addressing structural and non-structural measures are in place and covers monsoon contingency plan to address flood protection and early warning for climate extreme events [66]. The second criterion (AC-2.1) is based on three indicators covering aspects of agricultural rights in the context of diverse geographical areas, for example, restricting the rights for agricultural intensification in the highly susceptible ecosystem, including mountain slopes and dry river belts. The criterion (AC-2.1) also addressed the need for capacity development of relevant stakeholders including the local communities. The average index score (4.23) revealed that considerable efforts have been made to regulate farmers' rights. In the literature, it is opined that the regulation of rights, particularly land ownership, is important for the success of adaptive strategies as farmers opt to invest and enhance farming practices if they have ownership rights [67,68]. Besides that, the regulation of rights also impacts the farmers indirectly because many interventions, such as access to credit and crop insurance, are linked to land ownership.

Similarly, the average index score (6.61) for AC 3.1 i.e., climate-smart agriculture (CSA) practices shows considerable effort though the average index score (4.69) for climate infrastructure criterion (AC-7.1), suggesting the adoption of measures to substantiate the sustainability of CSA in the long term. Most commonly adopted CSA practices in Pakistan include changing fertilizer types and quantities, shifting cropping patterns in accordance with seasonal shifts, using different seed varieties e.g., drought-resistant seeds, and soil and water conservation [69]. However, for the purpose of AC 3.1, agriculture practices for risk recovery in case of crops' failures, capacity enhancement of all relevant actors, and creating mass awareness about measures for minimizing GHG emissions from

livestock and cropping were considered. Adoption of CSA practices directly depends on climate infrastructure, for example, research-based technological solutions, the need for transportation, storage, and processing of agricultural products, and awareness-raising and capacity enhancement of farmers. Such CSA also requires support from technological innovation (AC 4.1), climate organization (AC 5.1), and institutional effectiveness (AC 6.1) for which the average index scores were 5.40, 5.74, and 6.71, respectively. The functional linkages between these criteria affect the performance and sustainability of other criteria. For example, research and development are crucial to developing drought-resistant, pest-resistant crop varieties, and achieving irrigation efficiency, which will not only require research institutions but also extension services to disseminate the research [69].

The eighth criteria agreed for the study is very crucial as it addresses the water, agriculture, and energy nexus. These three sectors are most impacted by climate change [20,49,70]. The average index score against the AC 8.1 remained at 5.36 i.e., Fair. Agricultural practices influence the water and energy demand in Pakistan [71]. In the past, subsidizing water and energy to boost agriculture production has resulted in declining groundwater reserves, water logging, and salinity [72]. A poor understanding of the interlinkages of these sectors has triggered unsustainable use of resources and poses a serious risk to environmental security and sustainability [71]. The fact is evident from the poor index score (3.37) for the AC 9.1 criterion that deals with sustainability. Due to poor coordination among sectors, institutions often make their decisions in an isolated and fragmented way, which results in boosting one sector but with long-term implications for the resource base.

The responses of the stakeholders supported the assertions by identifying gaps in response strategies at the provincial and district levels in the agriculture sector of Pakistan, though steps taken at the federal level were compatible with climate. After the 18th Amendment of the Constitution in 2010, provinces are putting in their utmost efforts, which is evident from the index score (4.60) but they still lack regulations, and climate infrastructure. Consequently, the index score (1.91) for the criteria of sustainability remained significantly low. Major gaps exist at district levels where index scores are not encouraging across Pakistan. The main reasons are linked with the capacity issues at the district levels due to the institutional mechanism in a local governance context that could not flourish over a period of time. In the past, different local governance system designs were introduced. In the recent past, some Local Adaptation Plans of Actions (LAPAs) were also piloted in selected districts [73,74]. However, it always remained a challenge to ensure a vibrant system particularly for the agriculture segment. Relevant federal policies and programs cover all dimensions of CCD in agriculture and provide a plausible explanation that is commensurate with the index scores. "National Food Security Policy (NFSP), 2018" is focused and harmonized to address these issues and is coherent with "(FICCP)", 2014. NFSP, 2018 provides a commitment to building a climate resilient system for agriculture. Contrary to federal policies, most of the provinces, including Khyber Pakhtunkhwa, Balochistan, Azad Jammu Kashmir, and Gilgit Baltistan need to revise their existing policies. Punjab province has developed the "Punjab Agriculture Policy, 2018" that desires concrete short- and medium-term actions for its vision i.e., a "diversified, sustainable, modern and market-driven sector". It is coherent with FICCP. Similar to the province of Punjab, the "Sindh Agriculture Policy (2018–2030)" also addresses emerging challenges.

A criteria-wise comparison of the index scores revealed that the lowest score (3.37) is of criterion 9.1 i.e., sustainability, though it is fair at the federal level (5.30) but lies in the range of very poor to poor for the provinces (2.90) and districts (1.91). It was deduced that the federal government needs to play its role in strengthening the provincial and district departments for a coherent and inclusive governance framework. Often, climate change mitigation and adaptation are considered to be confined to a single level of governance or, if multiple levels are considered, those are viewed in the context of a top-down approach. The findings contradict the existing governance paradigm and clearly show that climate change concerns should be addressed at all levels from the local to the global. The interaction of the variables among governance levels is complex and

multi-faceted. For example, adaptation to climate change at the district level is crucial but water resource management for agriculture is the policy area that is essential for adaptation and needs to be supported by appropriate national policies. Therefore, the development of climate infrastructure (AC-7.1) to strengthen existing institutional capacities is crucial, for which agriculture sector strategies, policies, and planning documents do not provide a roadmap. The findings are evident from the relatively low governance index score (4.69) compared to other criteria except for sustainability. Another key factor in policy implementation is coordination among federal, provincial, and district departments. For example, policy measures for achieving irrigation efficiency, and small dams, etc. cannot be materialized without coordination with the provincial irrigation department. Coordination among departments is more crucial after the 18th Amendment in the 1973 Constitution. Responsibility for food security and agriculture is now shared by the Ministry of Food Security and Research at the federal level and the provincial agriculture departments.

Financial support is necessary for capacity building to adapt to climatic changes, particularly at the district level. Although agricultural departments are functioning, footprints are less evident due to a lack of clarity in mandates and consistency in policies. SDG-13 stresses "taking urgent actions to combat climate change and its impacts". The situation demands further research and the transfer of technology for the socio-economic uplift of farmers and building resilience capacity of the agriculture-dependent communities. However, the dearth of scientific knowledge compromises policy commitments, and the absence of a coordination mechanism to deal with climate change acts as a barrier in developing countries, such as Pakistan. Besides this, limited knowledge about available resources and capacity building is exacerbating the situation. The evaluations based upon the data from the grass-root levels are obligatory for the success of the designed efforts based on a top-down policy mechanism [75].

## 5. Conclusions

Scientific data on the causal relationship between anthropogenic activities and climate change poses a major challenge for decision-makers. Climate change is a daunting and perplexing hazard for global environmental security. The consequential impacts on the agro-based and vulnerable regions have a domino effect on the social and demographic fabrics of society. The economically and socially marginalized segments of society may be the worst victims of climate-propelled calamities. Effective strategies for climate compatibility, environmental security, and sustainability can moderate or lessen the impacts of changing climatic phenomena in many ways. In this context, the application of the analysis model proved well and validates that agriculture has a very strong nexus with the water and energy sectors due to their complex interdependence and interplay in terms of adaptation, resilience, mitigation, and low carbon development to cope with the agonizingly increasing effects of the changing climate. The findings revealed significant gaps at federal, provincial, and district levels in policies, legal and institutional strategies to step forward the climate agenda in Pakistan's agriculture sector. However, the inadequacy is not homogenous at all levels of governance. It is deduced that all nine criteria are impacting each other, however, the basis of the null hypothesis regarding the placement of inclusive and adequate climate response vis-à-vis the government's policies, legal instruments, strategies, and other institutional arrangements is that there is no such mechanism so far established or in existence that cannot be rejected for the overall case of the first governance component. It is construed that a coherent and inclusive response mechanism to address the impacts of climate change in the agriculture sector is absent. The overall situation is similar to that which has been reported for developing countries in the United Nations Report on SDGs 2020. It is deciphered that provincial climate strategies are required along with enhanced coordination and capacities for execution at all tiers of constituencies. Further, there is a concurrence of opinion on the need for research, informed decision-making, and improved governance. This will require a shift from the traditional top-down model to a more comprehensive participatory governance framework. Besides this, the dissemination of

information pertaining to innovations, relief measures, and capacity enhancement programs is obligatory for the integrity of social, economic, and ecological infrastructure prior to ensuring climate compatibility, environmental security, and sustainability in agriculture sector endeavors.

**Supplementary Materials:** The following supporting information can be downloaded at: https://www.mdpi.com/article/10.3390/su141811763/s1, File S1: principles & Indicators-GC1Agriculture-KJ-10May2. File S2: GC1-AgricultureCCD9Aug22.

**Author Contributions:** K.M.J.I. extracted and shaped the basic idea, methodology, results, discussion, and conclusion from his Ph.D. thesis. M.I.K. supervised the overall work, reviewed and edited the overall paper technically and scholarly. N.A. and S.A. helped to write the introduction, discussion and conclusion sections. A.A.S. and M.A.U.R.T. contributed to the discussion, logical conclusion, and facilitation of the submission to a journal. W.U. assisted in referencing, formatting and proofreading. All authors have read and agreed to the published version of the manuscript.

**Funding:** This research received no external funding.

**Institutional Review Board Statement:** The research topic and questionnaire for primary data collection through KIIs and FGDs were approved by the Institutional Ethical Review Committee of International Islamic University Islamabad (Certificate No. IIUI/ORIC/IERC/110-415 dated 15 February 2017).

**Informed Consent Statement:** Although it was not applicable in the context vis-à-vis nature of the study, prior consent was taken before conducting KIIs and FGDs.

**Data Availability Statement:** The data that support the finding of this study are available from the first and corresponding authors on request.

**Acknowledgments:** This paper stems from a broad Ph.D. research study of the lead author. The authors express their gratitude to all the reviewers who helped in improving the paper by sharing their meaningful comments and constructive suggestions anonymously.

**Conflicts of Interest:** The authors declare no conflict of interest.

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
