# Peer review of "Multi-Variable Governance Index Modeling of Government’s Policies, Legal and Institutional Strategies, and Management for Climate Compatible and Sustainable Agriculture Development"

_sustainability, doi:10.3390/su141811763_

Round 1
Author Response
Kindly find attached file and see our response in blue text.

Reviewer 2 Report
Τhe article seems to fullfil the essentials in order to be published. Regarding the part that deals with the climate consequence s in Pakistan proves to be particullarly significant for the case climate consequences, eaiming at the adjustment of techniques in similar situations.
Author Response
Dear Reviewer,
We would like to thank you for your valuable comments about our manuscript titled “Multi-variable governance index modelling of government’s policies, legal, institutional strategies and management for climate compatible and sustainable agriculture development” (ID: sustainability-1891104). The following is response to the comment.
Note: Prefix “C” means Comment, Prefix “R” means Response
C1: Τhe article seems to fullfil the essentials in order to be published. Regarding the part that deals with the climate consequences in Pakistan proves to be particularly significant for the case climate consequences, aiming at the adjustment of techniques in similar situations.
R1: We are grateful for your endorsement of the paper with highly encouraging remarks.
Reviewer 3 Report
1. All the tables and figures must be greatly improved.
2. All results have no significant difference analysis, which reduces the reliability of the results
3. At present, the article is only a qualitative description, lacking quantitative statistics
Author Response
Kindly find attached file to see our response in blue text.

Reviewer 4 Report
Review of: “Multi-variable governance index modelling of government’s policies, legal, institutional strategies and management for climate compatible and sustainable agriculture development”.
This manuscript assessed the adequacy of climate response policies, legal and other appropriate arrangements in place for agriculture governance by taking the high vulnerability case of Pakistan. An assessment model combining qualitative and quantitative data and employing rule-based and right-based governance approaches were adopted to collect, analyze and validate. The findings reveal heterogeneous gaps at federal, provincial, and district levels, and provincial climate strategies are required. In general, this manuscript is well written. However, I have some major concerns regarding the method about how to convert the principle, criteria, and indicators into scores. A better interpretation would help understand the results.
General comments:
I found myself having some difficulties following the methodological framework. I would suggest the authors reorganize the section as it is important to interpret the results. For instance, Line 188-208, if I understand correctly, the authors introduce the six principles and nine criteria in the PCI framework. The principles and criteria are introduced cross with each other, leading to a less clear structure and repeating text. I would suggest separating the introduction of P and C. Meanwhile, do the climate governance principles (CPs) in Line 203 refer to the same thing as the six novel principles in Line 188-189? If so, please add the acronym at the first presence.
What is the relationship between the PCI framework and the CCD conceptual framework? How do the four parts of the CCD conceptual framework connect to the CPs? Is it possible to indicate the four parts of the CCD conceptual framework in Figure 1? Are the CCD response criteria the same as (09) criteria mentioned above?
There are repeating sentences about the measures and provisions when introducing each criterion. Meanwhile, I am not quite following the difference between the CCD agenda and the CCD conceptual framework.
Specific comments:
L23: The effect of climate change is different over the globe depending on many factors. Suggest removing “agonizingly”.
L33 and L39: Please spell out the acronyms.
L327-330: What is the source of the input data?
L442-445: Please add a reference for this statement.
Author Response

(The authors gave the same response as above.)

Round 2
Reviewer 4 Report
I have checked the response letter and the revision carefully and found that all my questions/comments raised in the first round of review have been replied satisfactorily by the author and the corresponding changes were made in the revised manuscript.
I therefore would like to suggest the acceptance of the manuscript for publication.